# Fast Multi-UAV Path Planning for Optimal Area Coverage in Aerial Sensing Applications

**DOI:** 10.3390/s22062297

**Published:** 2022-03-16

**Authors:** Marco Andrés Luna, Mohammad Sadeq Ale Isaac, Ahmed Refaat Ragab, Pascual Campoy, Pablo Flores Peña, Martin Molina

**Affiliations:** 1Computer Vision and Aerial Robotics Group, Centre for Automation and Robotics, Universidad Politécnica de Madrid, 28040 Madrid, Spain; marco.lunaa@alumnos.upm.es (M.A.L.); mo.aleisaackhoueini@upm.es (M.S.A.I.); 2Drone-Hopper Company, 28919 Leganés, Spain; a.refaat@drone-hopper.com (A.R.R.); pablo.flores@drone-hopper.com (P.F.P.); 3Network Department, Faculty of Information Systems and Computer Science, October 6 University, Giza 12511, Egypt; 4Department of Electrical Engineering, University Carlos III of Madrid, 28919 Leganés, Spain; 5Department of Artificial Intelligence, Universidad Politécnica de Madrid, 28040 Madrid, Spain; martin.molina@upm.es

**Keywords:** multi-UAV, coverage path planning, aerial sensing, bin-packing problem

## Abstract

This paper deals with the problems and the solutions of fast coverage path planning (CPP) for multiple UAVs. Through this research, the problem is solved and analyzed with both a software framework and algorithm. The implemented algorithm generates a back-and-forth path based on the onboard sensor footprint. In addition, three methods are proposed for the individual path assignment: simple bin packing trajectory planner (SIMPLE-BINPAT); bin packing trajectory planner (BINPAT); and Powell optimized bin packing trajectory planner (POWELL-BINPAT). The three methods use heuristic algorithms, linear sum assignment, and minimization techniques to optimize the planning task. Furthermore, this approach is implemented with applicable software to be easily used by first responders such as police and firefighters. In addition, simulation and real-world experiments were performed using UAVs with RGB and thermal cameras. The results show that POWELL-BINPAT generates optimal UAV paths to complete the entire mission in minimum time. Furthermore, the computation time for the trajectory generation task decreases compared to other techniques in the literature. This research is part of a real project funded by the H2020 FASTER Project, with grant ID: 833507.

## 1. Introduction

Nowadays, the research and development of unmanned aerial vehicles (UAVs) and unmanned aerial systems (UASs) are constantly growing due to their notable characteristics, low cost, ability to integrate payload, and autonomous navigation. These features have made UAVs a powerful system used in civilian and military applications. In addition, multiple professional solutions are offered by single-UAV systems [1], and swarms and multi-agent techniques are a line of research that is increasingly gaining the interest of researchers in the last decade due to its high performance in terms of time efficiency, flexibility, and fault tolerance [2].

In contrast, current multi-UAV systems present several challenges in their control layers, communications, decision-making capacities, and practical applications, including those for whom the human operator closes the control loop [3]. In summary, multiple approaches study the dynamics and behavior of each entity inside the swarm, the communication with other entities, and the intelligence of the swarm. Additionally, other external factors were considered during the design stages, including legal restrictions and the complexity of the piloting, or safety problems [4]. As such, the system’s complexity tends to increase proportionally to many entities. Based on this, different methods in the state of the art address the trajectory planning problem for multiple UAVs based on GNNS navigation [5,6] and multi-UAV task allocation [7,8]. In this context, multi-agent coverage path planning (CPP) is a subfield of trajectory planning where the algorithms have to find the optimal paths of UAVs equipped with sensors of a limited footprint to cover the free workspace [9] and the optimal path allocation for each UAV. This technique is applicable for tasks such as inspection [10], precision agriculture[11,12], search and rescue [13], remote sensing [14], and others.

Through several researchers’ works, it was found that multiple approaches deal with the problem of CPP for UAVs in different ways, as stated in [15], where the authors distinguished between no decomposition, exact, and approximate cellular decomposition techniques. For the first method, a defined area is split into sub-areas to determine the optimal route that establishes the connection between them. Another example, in [16], the authors divided the free workspace into cells and took the center of each cell as a waypoint within a graph; they found the resulting path by graph optimization. In addition, the methods represented in [17,18] proposed an exact cellular decomposition method dividing the area into convex subregions and determining the optimal path based on the number of UAV turns. On the other hand, no decomposition methods consider the entire area for planning and include works such as [19] which proposed an energy-efficiency spiral pattern for CPP or [20] that applied the back-and-forth strategy.

In the multi-UAV system, the aforementioned techniques are applied with the additional challenge of task allocation. Some approaches such as [11,21] apply exact cellular decomposition methods; in the first case, the authors use auction-based algorithms to assign tasks and in the other, with a leader–follower approach. Proposals as in [22,23] address the problem using grid pattern map decomposition and linear programming methods to solve the graph generated by the possible trajectories and the agents. Others such as [24] use a column generation model to perform the coverage mission with multiple UAVs using energy constraints; only numerical simulations were collected in the results. More similarly to our work, in the proposal presented by [25], the authors divided the coverage area between the agents based on their single relative capabilities and used the back-and-forth path planning method to cover each partition; moreover, in [26], the authors performed the same path planning technique for the total area, generating a graph with the possible routes for each fixed-wing UAV; this graph has multiple constraints to optimize the flight time and ensure the safety of the agents; the solution is found with the graph optimization using the mixed integer linear programming method (MILP). In this context, the algorithm in [27] presents a similar approach for multirotor UAVs, changing the constraints and the optimization method to optimize the computation time.

The present proposal was developed within the H2020 project FASTER (First Responder Advanced Technologies for Safe and Efficient Emergency Response)  [28]; in which the company Drone Hopper must provide a fleet of UAVs to be used by first responders (FR) to perform multiple tasks such as mapping, monitoring, surveillance, or search and rescue. This framework proposes a new multi-UAV coverage path planning technique based on the heuristics of a bin packing problem combined with optimization methods to generate the individual routes for each UAV that minimizes the mission time. Due to its simplicity, this algorithm consumes less computational resources when compared with other proposals in the literature. Furthermore, it is defined in the system architecture, communications, hardware, and user interface for a real-time operation, considering the FR’s requirements.

The paper is structured as follows: Section 2 describes the problem definition statement; then, in the Methodology in Section 3, the system architecture, the algorithms for the multi-UAV CPP, and the software implementation are presented. The results and discussion for the simulation and real-flight operations are then shown in Section 4. Finally, in Section 5, the conclusions and future work are stated.

## 2. Problem Definition

Through this research, it is assumed that there is a group of *k* heterogeneous multirotor UAVs at the same altitude covering a polygonal convex area in the $I\;R^{2}$ space defined by a set of points *P*; all UAVs are supposed to take off at the same time from different positions. The sensor footprint is the main feature to be considered in the parameter setting of any mission, as shown in Figure 1.

To have similar measurements inside the scanned polygonal area, all UAVs must reach the same altitude and the space between lanes must be constant. However, these parameters can be configurable without affecting the system’s performance.

Additionally, to prevent collisions, each UAV will be assigned a different altitude to take off, navigate to the mission start point, and return to launch (RTL), as shown in Figure 2.

For this reason, the problem cannot be solved by dividing the area by the number of UAVs due to two main factors: the distance of each UAV from the starting point of the mission and the assigned altitude. As an additional safety constraint, the paths of each UAV will be continuous without interruptions between them. Additionally, each aircraft is equipped with the same onboard sensors (RGB or thermal cameras pointing down). Furthermore, the system interface will be user-friendly to allow the operator to connect the ground control station (GCS) to the group of UAVs, select the *P* points to cover a convex area in a given satellite map, and then launch the flight operation. Finally, the algorithms must optimize the calculation time to operate in emergency scenarios.

With these requirements, the main challenge is to develop a system to determine the paths for each UAV to cover the defined area and define its architecture to complete the mission in the minimum amount of time. In the following sections, the solutions proposed to deal with these problems are described.

## 3. Methodology

### 3.1. Architecture Proposal

A centralized architecture is proposed, where the central node assigns the task to each agent. In this case, the GCS computer performs all the calculations and uploads the waypoints to each UAV. For this, the robotic operating system (ROS) [29] framework and MAVROS package [30] are used. The system architecture is presented in Figure 3.

As clarified in Figure 3, each UAV has an onboard computer (${OB-PC}_{n}$) to send and receive data from the flight control unit ($FCU_{n}$) through serial communication (for hardware details, see Section 3.4). The onboard computer captures the FCU and video cameras ($video_{n}$) and sends them to the GCS. The communications between the GCS and the UAVs OB-PC in the UAVs are performed using the standard IEEE 802.11ax (commonly known as WiFi 6). During the operation, the GCS obtains the input area from the user and calculates the optimal set of waypoints for each UAV. The waypoints are sent to each OB-PC using the ROS framework, which decodes the information for the FCU.

### 3.2. Multi-UAV CPP Algorithm

#### 3.2.1. Area Decomposition

Based on previous works in the literature [26,27], it is clear that they used the back-and-forth strategy shown in [31] for the area decomposition. This method establishes the first step to find the optimal coverage direction. The authors in [31] proposed that this direction must be perpendicular to the shortest height of the polygon to obtain the smaller number of curves optimizing the rows number; additional parameters as spacing and row distances are calculated based on camera footprint [32]. In this context, given the field of view θ, the aspect ratio *r*, and the flight altitude *h*; the sides $A_{F}$ and $B_{F}$ of the footprint could be calculated through the following equations:(1)$$A_{F}=\frac{2\cdot h\cdot \tan(\theta /2)}{\sqrt{1+r^{2}}}$$
(2)$$B_{F}=\frac{2\cdot r\cdot h\cdot \tan(\theta /2)}{\sqrt{1+r^{2}}}$$

Unlike the aforementioned approaches that generate waypoints to cover the rows of the path, the newly implemented algorithm in this research will obtain waypoints generated for the entire route (similar to cell decomposition methods) to optimize the distribution of tasks between the agents, as shown in Figure 4.

Figure 4 was generated using open source Matplotlib libraries [33] and shows a comparison of the generated waypoints to be used in the algorithms presented in [26,27] and the one implemented through the present work. The reason for the difference is simple: while the calculation time in graph optimization algorithms increases exponentially with each added waypoint, in other heuristic methods, such as that presented in Section 3.2.2, the processing time is slightly affected. In addition to this, spacing and row distance are placed considering the desired overlapping; nevertheless, multiple GCS software as QGroundControl [34] allows the user to set them manually; however, it is not the aim of this study to determine their optimal values.

#### 3.2.2. Multi-UAV Routing

Given the set of waypoints of the previous Section 3.2.1, it is proposed that the bin packing trajectory planner (BINPAT) is a routing strategy. This algorithm has two stages, respectively, track packing and task assignment. Assuming a constant average velocity for each UAV, the mission time for *k* UAVs can be modeled as
(3)$$T_{k}=\sum_{i=1}^{N}\sum_{j=1}^{N}\frac{D_{ij}}{V_{ij}^{k}}M_{ij}^{k}+Hd^{k}$$
where:(4)$$Hd^{k}=\frac{h_{m}^{k}+\Delta h^{k}}{V_{a}^{k}}+\frac{h_{m}^{k}+\Delta h^{k}}{V_{d}^{k}}$$
same as
(5)$$Hd^{k}=\frac{\left(h_{m}^{k}+\Delta h^{k}\right)\left(V_{a}^{k}+V_{d}^{k}\right)}{V_{a}^{k}\cdot V_{d}^{k}}$$

Through Equation (Equation 3), $D_{ij}$ represents the distance cost of flying between two nodes and $V_{ij}^{k}$ represents the UAV flight speed. The binary variable $M_{ij}$ defines whether the $k_{th}$ UAV travels from point *i* to *j*; at this point, operational constraints proposed in Section 2 determine that all paths will be continuous. Finally, Hd (Equations (Equation 4) and (Equation 5)) is the delay caused by the altitude assignment and will depend on the mission altitude ($h_{m}$), ascent velocity ($V_{a}$), descent velocity ($V_{d}$), and the difference between assigned and mission altitudes ($\Delta h^{k}$). Consequently, the total mission time is the maximum $k_{th}$ UAV mission time. With these considerations, the optimization will be performed by minimizing the maximum $I\;R^{2}$ distance-based mission cost, that is: (6)$$C^{k}=\sum_{i}\sum_{j}D_{ij}M_{ij}^{k}+Hd^{k}$$

Based on this, the algorithm for the BINPAT is presented in Algorithm 1.
**Algorithm 1:** BINPAT Algorithm
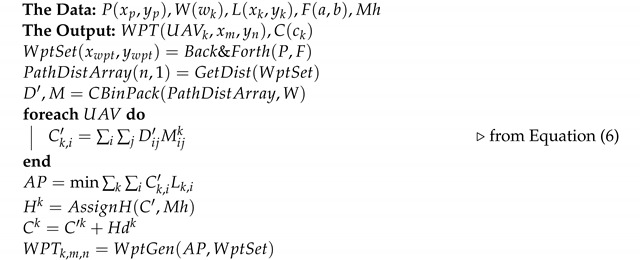


    In Algorithm 1, the inputs are given by the set of points (*P*), the UAV weighted parameters (*W*), UAV locations (*L*), the camera footprint parameters (*F*), and the mission altitude (Mh). The UAV weighted parameters vector represents weighted features that can influence the mission performance; these weights are the same when having a homogeneous multirotor. In addition, the outputs are the individual cost vector *C* and the three-dimensional matrix WPT with the set of 2D waypoints for each UAV. In the first stage of the algorithm, the set of waypoints is obtained using the back-and-forth (Back&Forth) technique (Section 3.2.1); then, the waypoints are separated into individual paths and each path distance is computed. During the track packing step, the result of the custom bin-packing algorithm (Algorithm 2) is the set of individual waypoints, where $D^{\prime }$ is the distance that a UAV has to navigate based on its initial weight and *M* represents the traveling condition (if a track is assigned to a UAV or not). For the task assignment stage, the asset cost matrix (AP) is obtained through the modification on the Jonker–Volgenant method for a linear sum assignment problem [35,36]. After that, the altitude assignment matrix ($H^{k}$) is obtained proportionally to the first cost matrix ($C^{\prime }$), i.e., UAVs with higher mission costs will be assigned lower takeoff altitudes. Then, the resultant cost vector (*C*) is calculated, adding the individual altitude delay cost ($Hd^{k}$) to the first cost matrix; finally, from AP and WptSet, it is calculated as WPT.

Furthermore, Algorithm 2 is an analogy to the bin-packing problem, in which a set of tracks (items) must be assigned to a set of UAVs (bins). In this case, the capacity of each UAV $W_{k}^{\prime }$ is the weighting of the total cost of the mission determined by the vector of weights *W*. Starting from mentioned values, tracks are iteratively assigned to each UAV until their maximum capacity is completed. As such, adding a new track to the UAV should not exceed its maximum capacity or improve the result of not adding it. As a result, the mission distance and the indices of the tracks to be traveled by each UAV are calculated.
**Algorithm 2:** CBinPack Algorithm
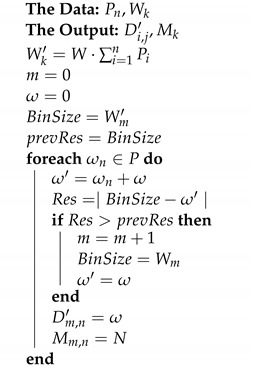


#### 3.2.3. Routing Optimization

Although the BINPAT algorithm can represent an optimal implementation result for specific UAVs (see Section 4), there may be cases where using fewer UAVs optimizes the mission time. BINPAT could not determine it since it will try to assign the entire mission to the set of available UAVs (especially for the case of homogeneous UAVs). For this reason, a new variation called Powell optimal BINPAT (POWELL-BINPAT) was implemented to solve the optimal assignment problem. In Section 3.2.2, one of the inputs of the BINPAT algorithm was a set of initial UAV weights $W_{k}$, and the output was the individual set of waypoints and the individual mission cost whereby the optimal mission cost will be found by minimizing the maximum individual cost (Equation (Equation 7)):(7)$$\min\left(\max C^{k}\right)$$

For this reason, the objective of this step was to find the best set of weights $W_{k}$ for *M* given UAVs that produce the optimal cost subject to Equation (Equation 8):(8)$$0\leq W_{k}\leq 1,\forall k\in M$$

Dealing with $W_{k}$ as the input and the result of the BINPAT algorithm $C^{k}$ from Equation (Equation 6), an iterative process is used to enhance the minimum cost of the mission, and this is achieved using the Powell optimization method [37,38] using as main parameters Equations (Equation 7) and (Equation 8), respectively. Powell’s technique finds the best solution by performing one-dimensional minimizations along each vector of the directions in the N-dimensional set of solutions.

### 3.3. Software Implementation

The novel swarm application is implemented in both the virtual and real world, wherein the former, all the algorithms are interpreted in C++ and Python codes and then compiled in the Debian space, connected to a Qt visual application, as shown in Figure 5, to be easily modifiable, interacting the generated data using ROS bridge platform, as displayed in Figure 6, and finally, simulated inside the Gazebo area to check whether the swarm drones display any incoherent action. This subsection extensively addresses the schematic of accomplished work through simulation.

The graphical user interface (GUI) environment designed in this research is based on Qt C++ language, interacting with the ROS workspace to regulate the swarm algorithms and communicate with the drones. Accordingly, Figure 5 shows the main elements of the GCS SW divided into four main sections, connection, streaming tabs, flight instruments, and the map widget. The GUI imports all the generated drones’ data, including drone ID, position, attitude angles, velocities, and their battery information transformed to corresponding **.json* files, as shown in Figure 5, and according to the user-chosen waypoints, a mainframe area for the swarm is created to be exported as **.json*, **.yaml*, and **.wp* formats to be further utilized by the main algorithms to generate the interpolate and extrapolate points close to the frame. Optimizing calculated routes, the best route (considering the equal distance and flight time, the one that performs the fewest deviations is the most optimized method) will be the candidate to pass the ROS bridge and be subscribed by the simulation application. Furthermore, Figure 5 demonstrates a sample swarm mission in a stream, starting in the upper-left part, where three connected drones are displayed and then the *PUBLISH ON COP* button starts advertising the online location of the drones and sends the data to the ROS workspace. While the upper-middle section relates to the swarm parameters, *Swarm Lines Distance* determines the gap between each line of the mission in *meters*, while *Swarm Altitude* defines the flight altitude of all drones during the mission (more information is given in Section 2). In addition, setting the swarm parameters, creating a set of waypoints on the map, and then *saving* the green button exports them through the algorithms. On the upper-right corner of Figure 5, however, all information received from the connected drones is monitored and each drone could be directly armed or be returned to the landing point near the *Home* location by using *ARM* and *RTL* buttons, respectively; moreover, the three flight instruments show the necessary information of the flight regarding each connected drone. Additionally, the middle blue windows contain the number of waypoints chosen by the user and their relative distances to the *Home*, the right part demonstrates that all waypoints after that are produced by the swarm algorithms and imported in the application. Eventually, the map section shows the online location of the drones and the generated waypoints. Mainly, since all the path planning and optimization codes are compatible with ROS workspace, they are modulated with ROS launchers to rapidly change the main variables, which adds **.yaml* formats and contains the addresses and properties of the connected drones which accelerate the running of the whole application. Furthermore, the SW is enriched with several **.qml* codes to enhance the map view of the mission area to make it as realistic as feasible to choose waypoints, view determined routes, and have an online drones view. Further SW details are collected in Table 1.

As mentioned in Table 1, four programming languages are utilized to ensure each part of the SW performs optimally; for instance, there are myriad types of primary flight displays (PFDs) that visualize the actual state of the drone inside the SW; however, a great number of PFDs decelerate the system because of the large volume of code compiling behind. Here, nonetheless, a pre-designed Qt-based collection, QFlightInstruments: http://marekcel.pl/qflightinstruments (accessed on 10 March 2022) [39] is employed that loads dramatically quickly (with 200 Hz frequency) and demonstrates images with a high resolution using image coding (H.264). Through Figure 6, a complete schematic diagram for the swarm mission procedure is implemented, showing the online hybrid map used in the GUI, the communication of the GCS, and the swarm drones through an ROS interaction that concludes in generating the swarm waypoints to output the mission.

The communication status of the drones can be checked at any time before starting the mission using *return-value ssh*; then, connected ones start publishing the data on the *Kafka* server confined by a local network to advertise the flight data for the GCS and other drones. Definitely, the distance between the swarm lines and the number of surrounding mission lines is configurable before starting the mission. Meanwhile, the movement trace of the drones and the streaming lines are demonstrated on the hybrid map area.

### 3.4. Hardware Implementation

To conduct practical missions, a set of system configurations is chosen to function optimally; meanwhile, the GUI application supports monitoring up to eight drones simultaneously. Clearly, each drone is equipped with an onboard computer to function as a slave system for the ground master computer, communicating through an ROS network connection. Furthermore, the unmanned aerial system (UAS) employed a robust wireless dual-band connection lay on a TP-link 4G+ Cat6 AC1200 Wireless Dual Band Gigabit Router which is then enhanced by connecting to the global network; i.e., establishing unique identification ports (IPs) for each drone to be in a local communication published on the global network. Moreover, the drone’s configuration is shown in Figure 7 where onboard PCs advertised ROS topics for the master and further listened through the GCS. In addition, a set of video streaming is configured using OpenCV libraries for flying drones in which the GUI software received a real video stream from each drone through the local network, as shown in Figure 3.

Ultimately, Figure 7 demonstrates the equipped equipment for the drones through the swarm mission where each drone is empowered by various components: a Pixhawk 2.1 CubePilot integrated with here3 GPS, a powerful onboard PC (Jetson Xavier Developer), which has an eight-core ARM processor based on a 64-bit CPU module, a ZED2 StereoLabs camera mounted below the Jetson Xavier, and a wireless ASUS Dual-Band (5 GHz/2.4 GHz) Wireless USB 3.0 Adapter communicating with the TP-link router. Overall, the system integration utilized through this project, according to several observations, aims to conquer multi-UAV path planning, communication problems, and internal uncertainties.

### 3.5. Transformation between Relative and Absolute Coordinates

To prevent unbiased errors through the transformation of coordinates from the SW, ROS bridge, and the drones, a common interpreted form is assumed, that is, relative coordination. As such, waypoints generated by the user are transformed from absolute location (latitude and longitude) to the relative distances from the pre-defined *Home* location and then exported to the ROS workspace, as shown in Equation (Equation 9). According to Haversine formula [40], which determines the orthodromic distances of two points on a sphere and considering the radius of the Earth (6371 km), Equation (Equation 9) could be driven as
(9)$$\begin{array}{c} \Delta lat=RAD(lat_{p}-lat_{h})\\ \Delta lng=RAD(lng_{p}-lng_{h})\\ d=\sin{\left(0.5\Delta lat\right)}^{2}+\cos\left(D2R\left(lat_{h}\right)\right)\cos\left(D2R\left(lat_{p}\right)\right)\sin{\left(0.5\Delta lng\right)}^{2}\\ \to \mathrm{Relative}\;\mathrm{Distance}=2R\arctan(\sqrt{d},\sqrt{1-d}) \end{array}$$
where *h* represents the *Home* location, *p* is the target waypoint; function RAD transforms an angle from degree to radian; and lat and lng represent the latitude and longitude of the waypoints, respectively. Using Equation (Equation 9) through the SW planning, all the location data are relative for both GUI and the swarm algorithms to be with the same scale.

## 4. Results and Discussion

### 4.1. Algorithm Validation

The algorithms described in the previous sections were implemented using Python libraries for validation. At the same time, numerical simulation was tested in multiple scenarios to evaluate different combinations before implementing the real flights. The main idea is to combine simple back-and-forth and sampled back-and-forth for area decomposition, as shown in Figure 4, with route generation algorithms (BINPAT and POWELL-BINPAT). In addition, three possible combinations are evaluated: BINPAT+simple back-and-forth (SIMPLE-BINPAT), BINPAT+ sampled back-and-forth, and POWELL-BINPAT + sampled back-and-forth. The distance-based cost results are represented in Table 2 with the graphical results shown in Figure 8.

According to Equation (Equation 7), the total cost of the mission will be the highest of the individual costs for each UAV. To clarify the results in Figure 8 and future figures, the navigation order of the routes is from the lowest to the highest index. Regarding Table 2 and Figure 8, it can be noticed that SIMPLE-BINPAT reports the greatest cost (963.81 m). In contrast, the sampled back-and-forth decomposition improves the performance, first for the BINPAT (928.75 m) and then for the POWELL-BINPAT (911.77 m). Additionally, as a result of the optimization, the POWELL-BINPAT reduces the standard deviation of the cost, meaning that each UAV consumes similar resources to perform the mission. Through the context, graphical comparisons with other proposals in the literature are represented, as shown in Figure 9.

In Figure 9, the results obtained with simple and sampled back-and-forth decomposition are compared with the graph optimization methods in the literature [26,27]. Although the constraints used for the optimization in these methods are different, the BINPAT algorithm with simple area decomposition generates similar trajectories compared to these proposals. Moreover, using the approaches with sampled area decomposition, these results are improved for our specific constraints. Furthermore, the processing time was evaluated in a computer with an AMD Ryzen 7 3750H processor and 16-GB RAM, as shown in Table 3.

Based on the results presented in Table 3, it can be noticed that BINPAT achieves acceptable results in computation time during the planning stage when it is compared with the other proposals for coverage routing that uses processors with similar features. For instance, in the case of three UAVs, BINPAT variations achieve between 0.0014 and 0.3 s; in contrast, the approach presented by [11] reported processing times between 0.15 and 0.31 s for three UAVs. Moreover, the algorithm presented in [26] achieves computation times from 0.8 to 4.59 seconds depending on the constraints; and through the work presented in [27], the authors stated a computation time of 0.8 seconds for three UAVs and 19 waypoints.

Finally, one of the main features of POWELL-BINPAT concerning BINPAT is its ability to decide the optimal number of UAVs needed to complete a mission in the shortest time. Through Figure 10 and Table 4, the optimum UAV assessment results and the comparison of the cost-based results as represented show the best results of planning a mission, in contrast to the work performed in [26,27].

Furthermore, by analyzing Table 4, it is clear that UAV1 generated unnecessary distance costs by climbing to the assigned altitude, reaching the first waypoint, and returning; these costs are optimized with the Powell method to obtain better results in only planning missions for UAV2 and UAV3.

### 4.2. Simulation Results

The presented approaches were tested in the open source Gazebo simulator [41] with PX4 software in the loop (SITL). Two simulation scenarios are considered: short-track scenario and long-track scenario, as shown in Figure 11.

The difference between these scenarios is the length of the lane; while on the short track, the maximum street length is approximately 150 m, on the long track, the maximum is approximately 320 m. Furthermore, the experiments were carried out with three and five UAVs. Multiple mission altitudes (from 35 to 55 m) and different lane widths (between 5 and 15 m) were tested. The main objective was to evaluate the influence of multiple parameters in the algorithm’s performance such as the number of UAVs, the altitude, and the lane features. As an additional detail, the cruise velocity of each UAV is considered 5 m/s, and the route was completed with a constant heading. The metrics used to evaluate the simulation results depend on the time that the UAVs take to accomplish a mission. They include the maximum time (Max), the average time (Av), the standard deviation (SD), and the coefficient of variation (CV), as shown in Equation (Equation 10):(10)$$CV=\frac{SD}{Av}*100$$

The tables with the time results of multiple experiments are presented in Appendix A. Time histograms for average UAV time results obtained from Table A1, Table A2, Table A3, Table A4, Table A5, Table A6, Table A7, Table A8, Table A9, Table A10, Table A11 and Table A12 are shown in Figure 12.

In Figure 12, POWELL-BINPAT achieves better results in the mission assignment task for three and five UAVs compared with the BINPAT and SIMPLE-BINPAT approaches. This fact can be noted by evaluating the coefficient of variation (CV) in each mission; in this context, the behavior of the coefficient of variation is shown in Figure 13.

In Figure 13, the CV indicates the uniformity in the distribution of the mission for each UAV. In general terms, the results obtained with the short-track scenario are lower compared with the values obtained for the long-track scenario. Moreover, the coefficient of variation is more sensitive to changes in the mission parameters and the number of UAVs in the SIMPLE-BINPAT method. Although the changes are milder in BINPAT; in POWELL-BINPAT, the values tend to be lower and more uniform in all cases. Finally, the average maximum time for each method is presented in Table 5.

In Table 5, SIMPLE-BINPAT has the lower maximum time in the short-track scenario; however, this is similar to the other approaches. The main differences can be noticed in the long-track scenario, where SIMPLE-BINPAT has a poor performance and POWELL-BINPAT is better than the other algorithms.

### 4.3. Real-Flight Test Results

The scenario for the real test results was similar to the short-track stated in Section 4.2. A picture of the pictures of the real implementation is shown in Figure 14 and the routes completed with the POWELL-BINPAT method obtained from the datalog are presented in Figure 15.

The altitude of the mission was 20 m and the space between lanes was 10 m; moreover, the cruise velocity was set to 5 m/s for each UAV.

Time results are presented in Table 6 and the time histograms are shown in Figure 16.

The implementation results show a behavior similar to the simulation in the Gazebo software. POWELL-BINPAT optimizes the total mission time by minimizing the difference in the individual flight time of each UAV. The real flight results are thus consistent with those obtained in multiple simulations.

Finally, an early implementation of BINPAT for two UAVs with simple area decomposition could be found at the Policía Municipal de Madrid: https://www.youtube.com/watch?v=m6CssNmgwH0 Youtube Channel (accessed on 10 March 2022).

## 5. Conclusions and Future Work

This paper presents a complete software framework, algorithms, and a special architecture for the practical implementation of area coverage missions for multiple UAVs. This system was generalized to be used with different sensors in function of the application, with the only possible variations being that of the sensor footprint (to calculate the lane width and waypoint separation).

The results of the validation tests show the performance of the algorithms in multiple hypothetical scenarios. The hypotheses are corroborated through the Gazebo simulator and during real-flight tests. The two main highlights of BINPAT and POWELL-BINPAT are the low computation time for multiple waypoint paths and the ability to optimize the mission times.

Additionally, dynamic re-planning tests are proposed; i.e., if one or more drones are lost during the mission, the system will detect the missing waypoints and reorganize all the routes for the available UAVs.

As confirmed, the success of the algorithms largely depends on the correct calculation of the cost matrices; in this case, it has been tested with routes generated in the 2D space at a constant altitude. For future development, various 2D approaches and 3D trajectories could be analyzed (containing altitude changes along the path).

## Figures and Tables

**Figure 1 sensors-22-02297-f001:**
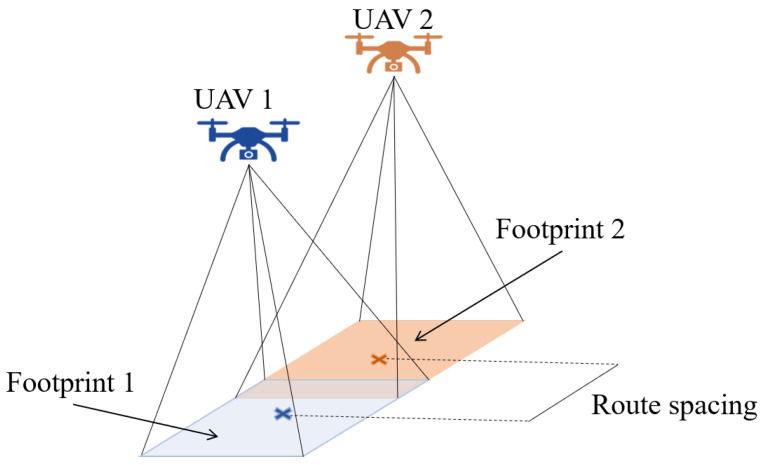
Footprint schema in a homogeneous UAV fleet.

**Figure 2 sensors-22-02297-f002:**
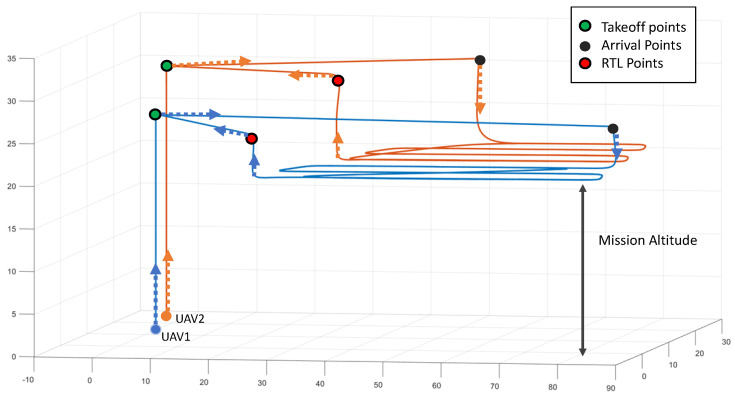
Mission operation schema. Each UAV takeoff at its assigned altitude, then performing the operation at mission altitude, and RTL at assigned altitude again.

**Figure 3 sensors-22-02297-f003:**
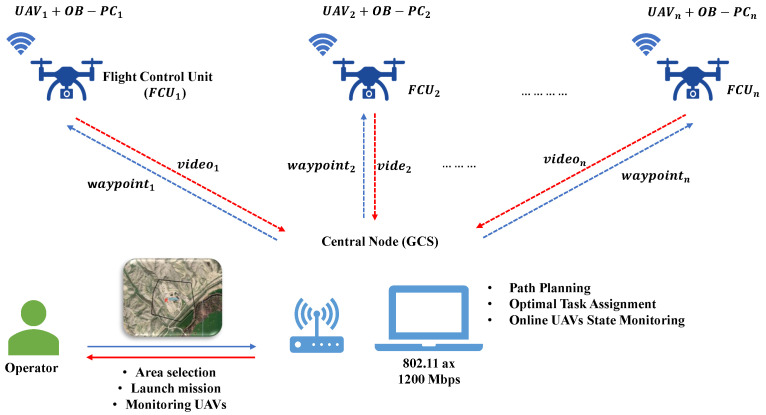
Architecture proposal.

**Figure 4 sensors-22-02297-f004:**
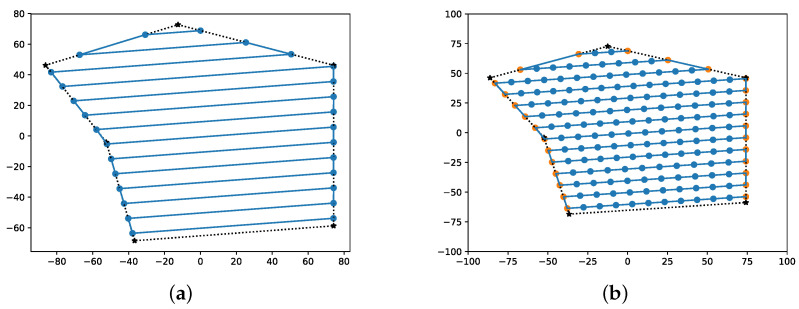
Waypoint generation results: (**a**) waypoints generated using simple back-and-forth algorithm; (**b**) waypoints generated with a sampled back-and-forth for our proposal.

**Figure 5 sensors-22-02297-f005:**
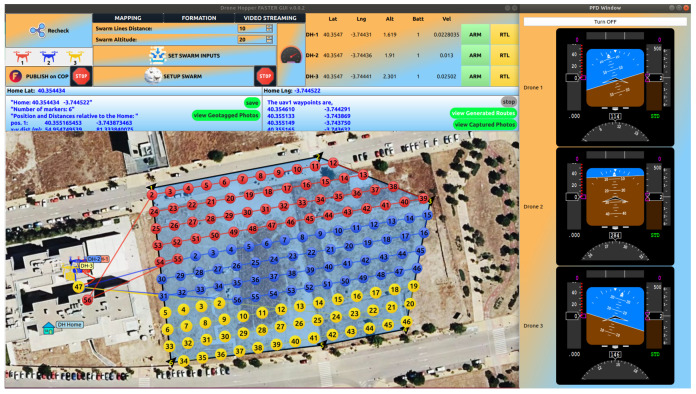
A snapshot of the ground control station (GCS) SW used for the swarm project.

**Figure 6 sensors-22-02297-f006:**
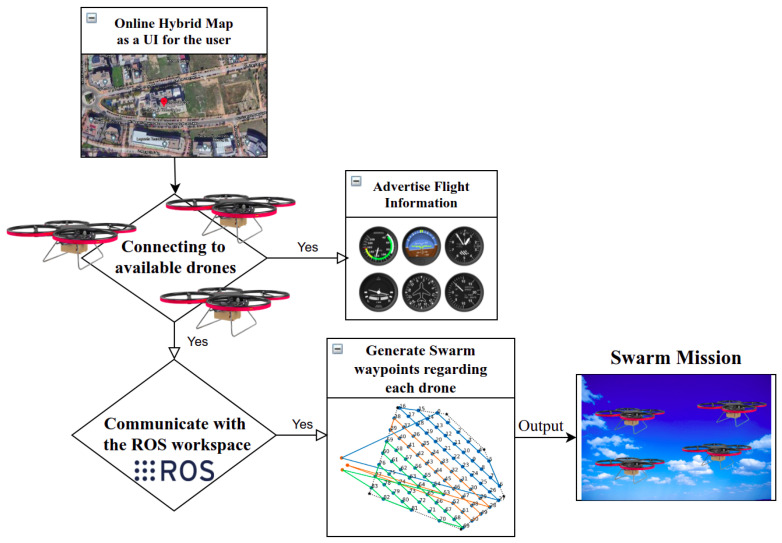
A diagram of the swarm mission procedure.

**Figure 7 sensors-22-02297-f007:**
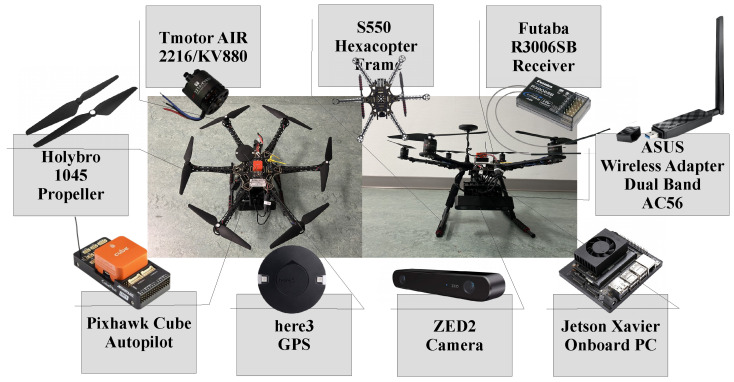
The drone properties utilized through the swarm mission.

**Figure 8 sensors-22-02297-f008:**
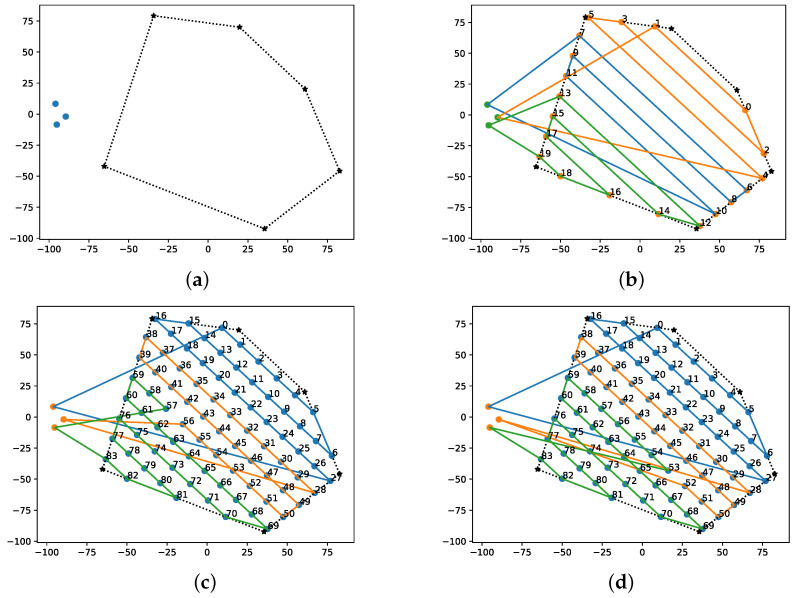
Multi-UAV algorithm results: (**a**) defined search area and UAV locations; (**b**) SIMPLE-BINPAT; (**c**) BINPAT; and (**d**) POWELL-BINPAT.

**Figure 9 sensors-22-02297-f009:**
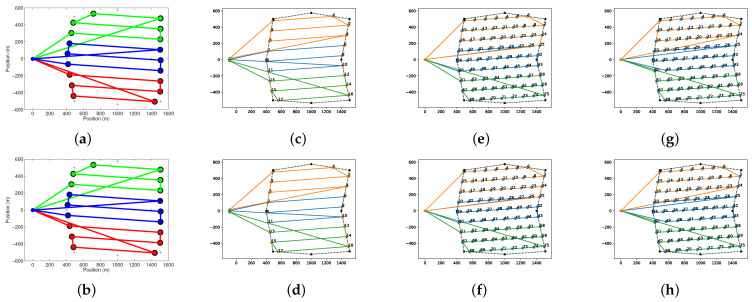
Multi-UAV routing comparison results with 3 UAVs: (**a**) results obtained with graph optimization for fixed wings; and (**b**) results obtained with graph optimization for multirotors; (**c**,**d**) SIMPLE-BINPAT; (**e**,**f**) BINPAT; and (**g**,**h**) POWELL-BINPAT.

**Figure 10 sensors-22-02297-f010:**
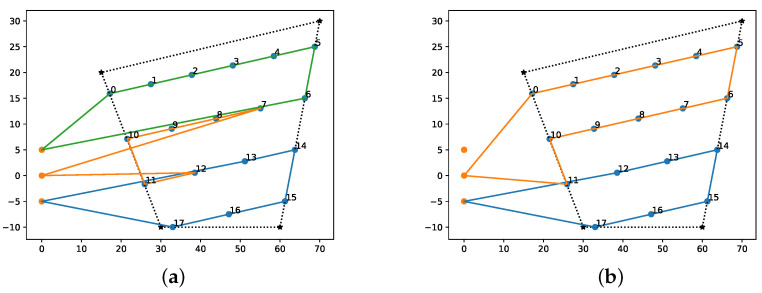
Optimal UAV assignment results with POWELL-BINPAT: (**a**) paths generated with BINPAT; and (**b**) paths generated with POWELL-BINPAT.

**Figure 11 sensors-22-02297-f011:**
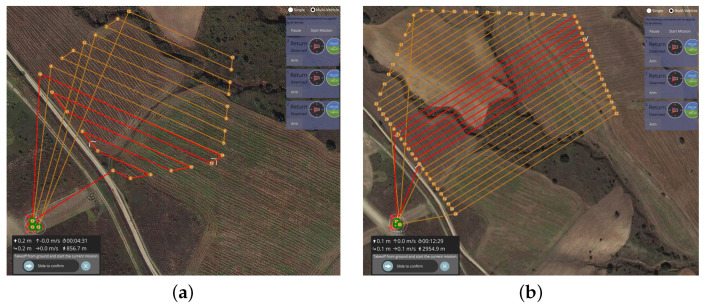
Robotic simulation scenario: (**a**) short-tracks scenario; and (**b**) long-tracks scenario.

**Figure 12 sensors-22-02297-f012:**
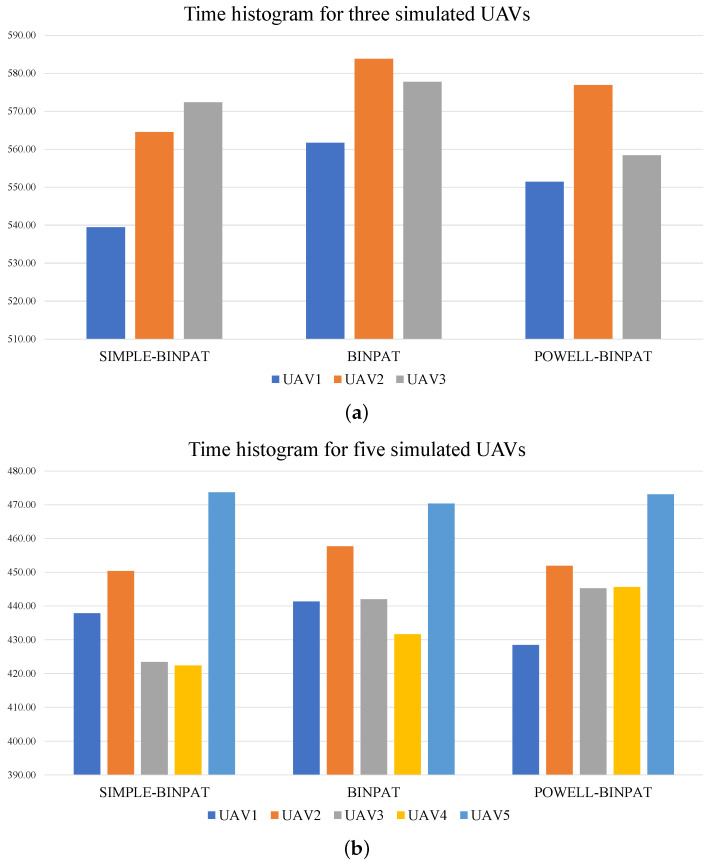
Time histograms for average results in simulation tests: (**a**) three UAVs’ time histogram; and (**b**) five UAVs’ time histogram.

**Figure 13 sensors-22-02297-f013:**
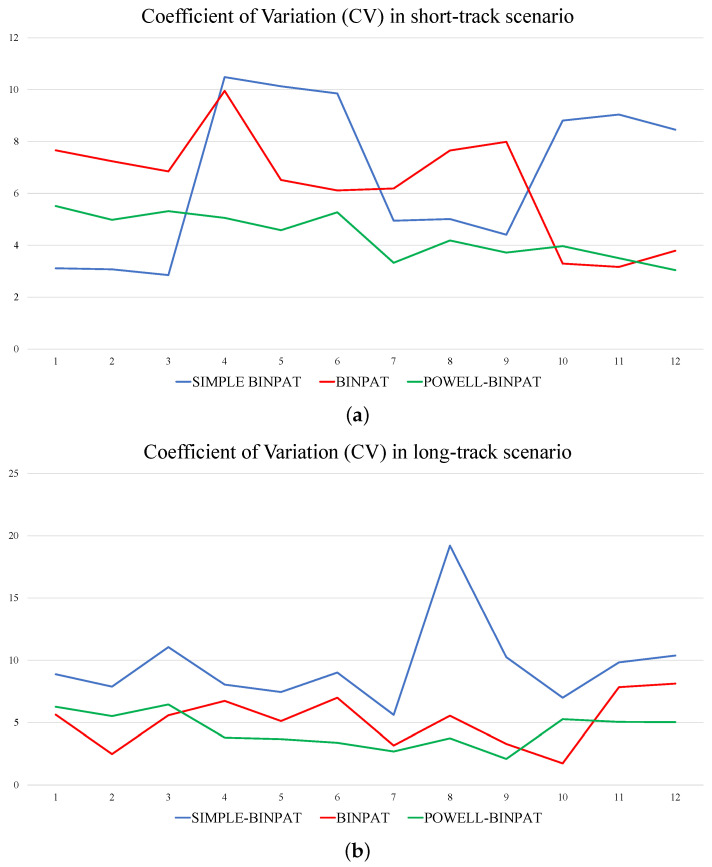
CV behavior obtained from experiments reported in Appendix A: (**a**) short-track scenario results; and (**b**) long-track scenario results.

**Figure 14 sensors-22-02297-f014:**
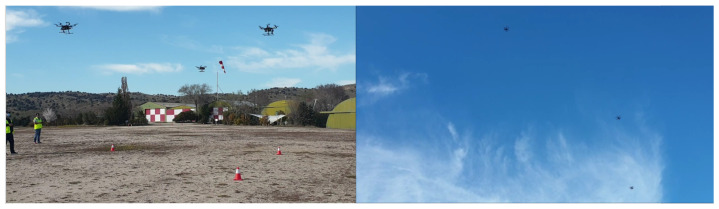
Pictures of the real implementation.

**Figure 15 sensors-22-02297-f015:**
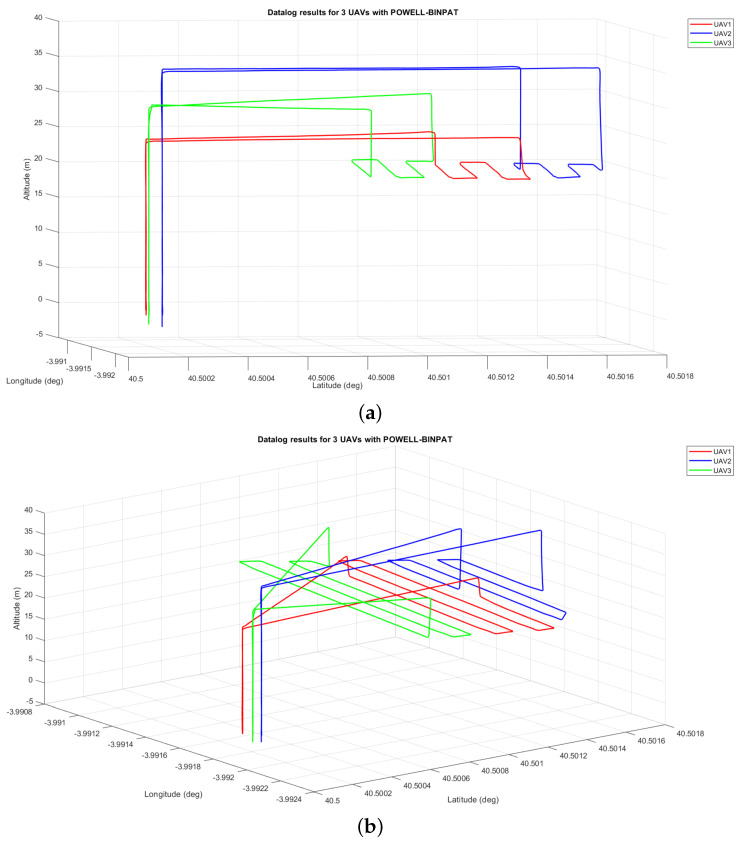
UAV positions reported by datalog in 3 UAV flights using POWELL-BINPAT: (**a**) main view; and (**b**) side view.

**Figure 16 sensors-22-02297-f016:**
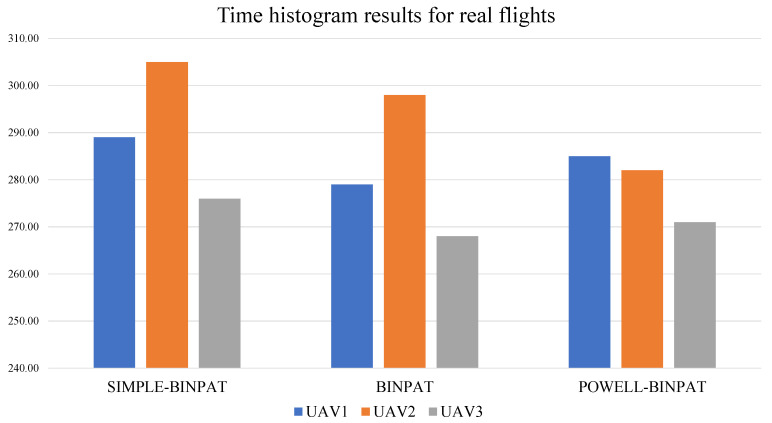
Time histogram for real flight with 3 UAVs.

**Table 1 sensors-22-02297-t001:** Full properties of the swarm GUI application.

Based Language	Role	Configuration
* **Qt C++** *	Contains the main functions, including widgets, buttons, frames, tabs, spinners, and text boxes; Connects the master computer to the drones as slaves, using ROS connection, utilizing **.bash* script handlers; Connects the main c++ file with previously generated files to moderate the swarm mission, linked to corresponding modifier buttons; Creates submodules to interact with **.qml* visualizers.	* **C++ 11** * * **Compiler qmake 3.0** *
* **Qt Meta** * * **Language (qml)** *	Contains visualizer objects, namely rectangles, circles, buttons, map-inputs, map quick items, map routes, map polygons, map poly-lines, map routes, list models, grids, and mouse functions; Manages the waypoint and the markers; Corresponding functions to save the user desired frame and view the calculated points; Determines the relative distances of the chosen points and the defined home which affects all the swarm algorithms.	* **Qt Quick** * * **JavaScript** *
* **Hyper Test** * * **Markup Language** *	Includes all the instruments with a high frequency, 100 Hz to update the flight information, related to the position, attitude (Euler angles), airspeed, altitude, battery, etc.; Streams the video of each drone during the flight, broadcasting through a local or global web address; Designates an especial API key of the Bing Maps for the map space used in the application;	* **HTML** *
* **Brian Fox Unix** * * **Shell (Bash)** *	Connects the drones to the GUI application via ssh connection and regulates the vulnerability when various drones are simultaneously connected; Manages the camera function of any connected drones, interacting with OpenCV libraries and fswebcam capturing SW; Launches or kills programs related to the swarm and optimization; Reports the online connectivity state of any drones during the flight to facilitate the ROS bridge to advertise and subscribe the topics according to available vehicles.	

**Table 2 sensors-22-02297-t002:** Distance-based cost results using the BINPAT and POWELL-BINPAT for simple and sampled area decomposition.

Distance-Based Cost			
**UAV**	**Simple Back-and-Forth**	**Sampled Back-and-Forth**	
	**SIMPLE-BINPAT**	**BINPAT**	**POWELL-BINPAT**
UAV 1	963.81	928.75	908.75
UAV 2	944.65	900.34	911.77
UAV 3	644.58	789.27	904.72
SD	179.03	73.7	3.53

**Table 3 sensors-22-02297-t003:** Processing time results for a different UAV number.

UAVs	SIMPLE-BINPAT	BINPAT	POWELL-BINPAT
2	0.000748	0.0021	0.296894
3	0.001474	0.002244	0.301134
4	0.001522	0.002756	0.512768
5	0.001961	0.003125	1.419654
6	0.003479	0.003365	1.853162
7	0.004275	0.003924	1.96757

**Table 4 sensors-22-02297-t004:** Cost-based results comparison.

Distance-Based Cost Results		
**UAV Number**	**BINPAT**	**POWELL-BINPAT**
UAV1	390.72	0
UAV2	401.93	370.73
UAV3	394.62	383.99

**Table 5 sensors-22-02297-t005:** Average maximum time for the simulation experiments.

	SIMULATION SCENARIO	
	**Short Track**	**Long Track**
**SIMPLE-BINPAT **	300.42	795.58
**BINPAT**	330.08	754.25
**POWEL-BINPAT**	322.00	735.42

**Table 6 sensors-22-02297-t006:** Time results for a real flight with three UAVs.

UAV	METHOD		
	**SIMPLE-BINPAT**	**BINPAT**	**POWELL-BINPAT**
**UAV1**	289.00	279.00	285.00
**UAV2**	305.00	298.00	282.00
**UAV3**	276.00	268.00	271.00
**Max**	305.00	298.00	285.00
**Av**	290.00	281.67	279.33
**SD**	14.53	15.18	7.37
**CV**	5.01	5.39	2.64

## Data Availability

Not available.

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
