# Peer review of "Fast Multi-UAV Path Planning for Optimal Area Coverage in Aerial Sensing Applications"

_sensors, 2022, doi:10.3390/s22062297_

Round 1
Reviewer 1 Report
The authors have proposed a complete framework for multiple UAVs area coverage missions planning. The proposed framework includes the algorithm for generating a back-and-forth path based on the onboard sensor data. Also, the authors have proposed two algorithms for individual path assignments - Bin Packing Trajectory Planner and Powell Optimized Bin Packing Trajectory Planner.
The paper is well-structured. It starts from the related research works review, then the original solution is proposed and experimentally validated. The experiments were carried out with the use of open-source robot simulator as well as real-world UAVs.
The following issues should be addressed:
1. Please clarify whether the experimental results of each UAV set come from many experiments or just from one experiment. It can be seen that some average values are presented in the tables, but it is not clear whether these are averages from 3/5 UAVs or from many experiments carried out with each set of UAVs. Of course, it would be better to carry out many experiments with each set of drones to have reliable results (the data should also be statistically analyzed in such a case).
2. The images presented in Figure 12 and especially 13 are too small.
3. Table 7 presents results for 3 UAVs, not 5 as stated in the caption.
4. Please improve the English language because there are grammar and style errors.
Reviewer 2 Report
- A key assumption is the mission will be performed at the same altitude for every UAV. This assumption complicates the problem. The authors should provide more motivation on this assumption and maybe they should also discuss the same problem when the altitude can be different for every UAV.
- The simulation setup should be presented in more detail, e.g., what is the altitude for short-track scenario and long-track scenario.
- It would be interesting to see more results showing how the altitude influences the performance (in terms of time or distance)
- Title of Table 7 is “Time results in seconds for five UAVs”, while there are results of 3 UAVs in the table.
Round 2
Reviewer 1 Report
The authors have addressed the most important issues, so in my opinion the paper can be accepted.
Reviewer 2 Report
The Authors addressed my all remarks in a satisfactory way, so I propose to accept the paper with no changes.